# Adipose tissue and skeletal muscle wasting precede clinical diagnosis of pancreatic cancer

Ana Babic [1,17], Michael H. Rosenthal [2,17], Tilak K. Sundaresan[3,17], Natalia Khalaf[4], Valerie Lee[5], Lauren K. Brais[1], Maureen Loftus[1], Leah Caplan [1], Sarah Denning[1], Anamol Gurung[1], Joanna Harrod[1], Khoschy Schawkat[2], Chen Yuan[1], Qiao-Li Wang[1,6], Alice A. Lee[7], Leah H. Biller[1,8], Matthew B. Yurgelun [1,8], Kimmie Ng [1,8], Jonathan A. Nowak[9], Andrew J. Aguirre [1,8], Sangeeta N. Bhatia [8,10,11,12,13,14,15], Matthew G. Vander Heiden [1,11,13,16], Stephen K. Van Den Eeden [5,18], Bette J. Caan[5,18] & Brian M. Wolpin [1,8,18] ✉

Patients with pancreatic cancer commonly develop weight loss and muscle wasting. Whether adipose tissue and skeletal muscle losses begin before diagnosis and the potential utility of such losses for earlier cancer detection are not well understood. We quantify skeletal muscle and adipose tissue areas from computed tomography (CT) imaging obtained 2 months to 5 years before cancer diagnosis in 714 pancreatic cancer cases and 1748 matched controls. Adipose tissue loss is identified up to 6 months, and skeletal muscle wasting is identified up to 18 months before the clinical diagnosis of pancreatic cancer and is not present in the matched control population. Tissue losses are of similar magnitude in cases diagnosed with localized compared with metastatic disease and are not correlated with at-diagnosis circulating levels of CA19-9. Skeletal muscle wasting occurs in the 1–2 years before pancreatic cancer diagnosis and may signal an upcoming diagnosis of pancreatic cancer.

Pancreatic cancer is currently the third most common cause of cancer death in the United States[1]. Due to lack of specific early symptoms and no clinically available biomarkers for early disease detection, more than 80% of patients are diagnosed after the tumor has spread beyond the pancreas when curative treatment attempts are rarely feasible[2]. New screening strategies and biomarkers of early pancreatic cancer are greatly needed to reduce mortality from this highly aggressive disease.

Pancreatic cancer is associated with multiple changes to systemic metabolism[3]. Many patients report weight loss, and a subset of patients develops diabetes within 3–4 years before diagnosis[4–8]. Furthermore, patients with advanced pancreatic cancer commonly develop cancer-associated cachexia, a wasting syndrome that leads

to muscle loss, functional impairment, and poor tolerance of anti-cancer therapies[9]. Alterations in circulating metabolites can be detected in patient plasma several years before a diagnosis of pancreatic cancer[10]. In mouse models of pancreatic cancer, changes in metabolite levels were accompanied by skeletal muscle and adipose tissue wasting that occurred early in pancreatic cancer development[10]. Furthermore, tissue wasting in these model systems was due in part to altered exocrine pancreatic function that developed with early pancreatic cancer, leading to nutrient malabsorption and wasting of peripheral tissues[11].

Among patients with localized pancreatic cancer, more than half of patients have pancreatic exocrine insufficiency (PEI) at the time of diagnosis as measured by fecal elastase testing[12], and a previous study

suggested that tissue wasting may be detected on CT imaging prior to a pancreatic cancer diagnosis[13]. Understanding the onset and trajectory of skeletal muscle and adipose tissue changes relative to pancreatic cancer development could have important implications for earlier detection in high-risk populations, as tissue wasting might serve as a biomarker of early, subclinical disease.

Using an automated machine learning algorithm to interrogate prediagnosis CT scans and an approximately 12,000 patient age-, sex-, and race-standardized CT reference set, we quantify body composition up to 5 years before diagnosis (Fig. 1), and we show that skeletal muscle wasting can be detected up to 18 months and adipose tissue wasting up to 6 months before a pancreatic cancer diagnosis.

## Results

### Patient characteristics

Characteristics of pancreatic cancer patients and matched controls at the time of prediagnosis imaging are shown in Table 1. Cases were more likely than controls to have a history of diabetes and smoke cigarettes. Pain (abdominal, back, flank, or chest) was the most frequent indication for the prediagnosis CT scan for both cases and controls (51 and 53%, respectively), and the overall distribution of imaging indication was similar between the two groups. Concordant with pancreatic cancer patients in the general US population[14], approximately half of the cases had distant metastatic disease at the time of diagnosis, and cases diagnosed with metastases had a substantially shorter median overall survival of 6 compared to 17 months for those with non-metastatic disease. The mean (SD) time between prediagnosis CT imaging and histopathological diagnosis of pancreatic cancer was 23 (17) months. For cases with an available at-diagnosis scan, the mean (SD) time between at-diagnosis CT imaging and histopathological diagnosis of pancreatic cancer was 10 (14) days. Subject characteristics by study site are shown in Supplementary Table 1.

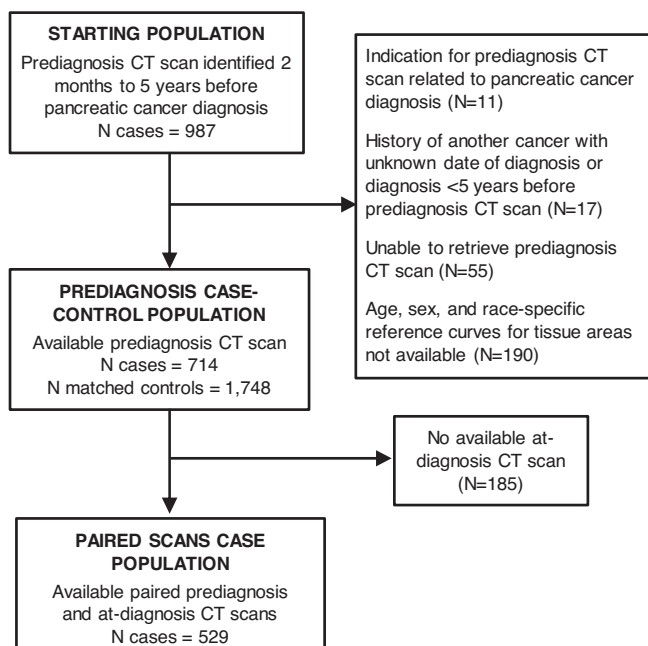

**Fig. 1 | Flow diagram of participants from matched pancreatic cancer case-control population.** *Age-, sex-, and race-standardized reference curves for skeletal muscle and adipose tissue areas were previously developed for white non-Hispanic females and males (18–90 years), black females (18–75 years), and black male (18–70 years) participants[26]. Age-, sex-, and race-standardized tissue percentiles could not be calculated from CT scans of participants outside of these categories. CT computed tomography.

### Skeletal muscle and adipose tissue in pancreatic cancer cases in prediagnosis period and in matched controls

To evaluate tissue changes in the period preceding diagnosis of pancreatic cancer, we compared age-, sex-, and race-standardized tissue percentiles between case and control populations at each of the four time intervals (>36 to 60 months, >18 to 36 months, >6 to 18 months, and 2 to 6 months before diagnosis). In the earliest time window studied (>36 to 60 months before diagnosis), cases had higher percentiles of skeletal muscle ($P = 0.001$), visceral adipose tissue ($P = 0.03$), and subcutaneous adipose tissue ($P = 0.02$) than matched controls (Fig. 2A). With the approaching pancreatic cancer diagnosis, these relationships inverted for subcutaneous adipose tissue and skeletal muscle, and percentiles for these tissue compartments were lower in cases compared to controls ($P = 0.04$ and $P = 0.03$, respectively) at 2 to 6 months before diagnosis. In contrast, BMI was not significantly different between cases and controls at either >36 to 60 months or 2 to 6 months before diagnosis ($P = 0.19$ and $P = 0.24$, respectively, Fig. 3A).

We next examined tissue measurements by time before pancreatic cancer diagnosis separately in cases and controls. Visceral and subcutaneous adipose tissue percentiles and BMI among cases with CT scans performed 2 to 6 months before diagnosis were significantly lower compared to cases with CT scans performed >36 to 60 months before diagnosis (all $P \leq 0.02$; Figs. 2B, 3B). In contrast, skeletal muscle percentiles were significantly lower in cases with CT scans performed >6 to 18 months ($P = 0.04$) and 2 to 6 months ($P = 0.002$) before pancreatic cancer diagnosis compared to cases with CT scans >36 to 60 months before diagnosis (Fig. 2B). For matched controls, no differences in adipose tissue percentiles, skeletal muscle percentiles, or BMI were observed for controls in the same time intervals compared to the controls within the >36 to 60 months interval (all $P > 0.16$, Figs. 2C, 3C). Although serial measurements were not available within individual cases, by examining age-, sex-, and race-standardized percentiles at the population level, skeletal muscle wasting appeared to develop earliest, with this wasting preceding the loss of weight and adipose tissues that developed closer to the time of cancer diagnosis.

We next evaluated the degree of weight and tissue loss in the time preceding pancreatic cancer diagnosis by examining changes from the prediagnosis to at-diagnosis CT scans among 529 pancreatic cancer cases with both imaging studies. During the mean (SD) time of 23 (17) months between the two scans, we observed a significant decrease in all three tissue compartments (all $P < 0.0001$; Table 2). Over this period, patients lost a median (percent) of 20.3 cm² (11.5%) of visceral adipose tissue, 20.9 cm² (11.4%) of subcutaneous adipose tissue, and 8.6 cm² (5.9%) of skeletal muscle. Among cases with available weight measurements, cases lost a median of 5.2 kg (6.3%), and their BMI declined by a median of 1.8 kg/m² (6.3%).

### Correlates of prediagnosis tissue losses in pancreatic cancer patients

We next examined the variability in tissue changes in the prediagnosis period and potential determinants of such changes (Fig. 4). The large majority of cases experienced tissue loss; in total, 73% lost skeletal muscle area, 71% lost subcutaneous adipose tissue area, and 70% lost visceral adipose tissue area. To evaluate whether the identified tissue wasting in the prediagnosis period was dependent upon the extent of cancer at diagnosis, we compared tissue wasting among cases with metastatic disease to those with a tumor localized to the pancreas. Notably, no differences were identified for changes in tissue measurements by tumor stage (all $P > 0.3$; Fig. 4), indicating that tissue wasting occurred to a similar degree in cases with localized and metastatic pancreatic cancer. To further evaluate the association between disease burden and tissue loss, we evaluated body composition change in the prediagnosis period in a subset of patients ($N = 135$) with available CA19-9 levels measured at diagnosis, as CA19-9 is known to be associated with overall disease burden[15,16]. We observed no

**Table 1 | Characteristics of pancreatic cancer cases and matched controls at the time of prediagnosis CT imaging**

| Characteristic[a] | Cases (N = 714) | Controls (N = 1748) | P value |
|---|---|---|---|
| Age, years[b] | 67.1 (10.2) | 66.8 (10.3) | 0.49 |
| Year of CT scan[b] | | | 0.39 |
| 2003–2010 | 314 (44) | 736 (42) | |
| 2011–2017 | 400 (56) | 1012 (58) | |
| Intravenous contrast with CT imaging[b] | 563 (79) | 1409 (81) | 0.32 |
| Female[b] | 340 (48) | 827 (47) | 0.89 |
| Race[b] | | | 0.63 |
| White | 659 (92) | 1603 (92) | |
| Black | 55 (8) | 145 (8) | |
| Study site[b] | | | 0.53 |
| DFCI/MGB | 228 (32) | 581 (33) | |
| KPNC | 486 (68) | 1167 (67) | |
| Body mass index, kg/m²[c] | 29.3 (6.1) | 28.9 (6.5) | 0.11 |
| Diabetes | 193 (27) | 368 (21) | 0.001 |
| Cigarette Smoking | | | <0.0001 |
| Never | 276 (39) | 772 (44) | |
| Past | 242 (34) | 674 (39) | |
| Current | 119 (17) | 199 (11) | |
| Unknown | 77 (11) | 103 (6) | |
| History of alcohol use | | | 0.44 |
| No/Unknown | 638 (89) | 1591 (91) | |
| Yes | 76 (11) | 157 (9) | |
| Prior personal history of cancer | 109 (15) | 177 (10) | 0.0003 |
| Time from prior cancer to prediagnosis scan, years | 12.7 (7.8) | 11.0 (6.6) | 0.06 |
| Indication for CT imaging | | | |
| Symptoms/suspected conditions of the GI system | 108 (15) | 189 (11) | 0.003 |
| Symptoms/suspected conditions unrelated to the GI system | 164 (23) | 548 (31) | <0.0001 |
| Systemic symptoms | 65 (9) | 160 (9) | 0.97 |
| New-onset diabetes | 4 (1) | 6 (0.3) | 0.44 |
| Pain | 365 (51) | 922 (53) | 0.46 |
| Other/unknown | 190 (27) | 361 (21) | 0.001 |
| Age at pancreatic cancer diagnosis, years | 69.4 (10.3) | NA | NA |
| Pancreatic cancer diagnosis period | | | NA |
| 2004–2008 | 108 (15) | NA | |
| 2009–2013 | 281 (39) | NA | |
| 2014–2018 | 325 (46) | NA | |
| Pancreatic cancer stage at diagnosis | | | NA |
| Non-metastatic | 370 (52) | | |
| Metastatic | 326 (46) | NA | |
| Unknown | 18 (3) | NA | |
| Mean time from prediagnosis CT imaging to pathological diagnosis, months | 22.7 (17.2) | NA | NA |
| Mean time from at-diagnosis CT imaging to pathological diagnosis, days[d] | 10 (14) | NA | NA |

*P* values were calculated using chi-square tests for categorical, and two-sided Wilcoxon rank-sum test for continuous variables.

*CT* computed tomography, *DFCI/MGB* Dana–Farber Cancer Institute/Mass General Brigham, *GI* gastrointestinal, *KPNC* Kaiser Permanente Northern California, *NA* not applicable.

[a]Characteristics obtained at the time of the prediagnosis CT imaging study in cases and matched controls. Mean (standard deviation) for continuous variables and *N* (%) for categorical variables, unless noted otherwise.

[b]Matching factors.

[c]Among 518 cases and 1628 matched controls with available body mass index at the time of prediagnosis CT imaging.

[d]Among 529 cases with available at-diagnosis CT imaging.

significant correlations of CA19-9 levels at diagnosis with changes in tissue measurements (all *P* values ≥0.1, Supplementary Fig. 1).

We next considered whether tumor location within the pancreas might be associated with these metrics. In particular, we hypothesized that a tumor in the head of the pancreas may cause greater pancreatic ductal obstruction and resultant altered pancreatic exocrine function, such that pancreatic head tumors would be associated with greater tissue wasting. However, we observed no difference in the amount of adipose tissue or skeletal muscle loss between patients with tumors in the head of the pancreas compared to the body and tail (all *P* ≥ 0.18; Fig. 4). We and others have previously shown that pancreatic cancer patients presenting with diabetes at the time of diagnosis experience greater weight loss in the prediagnosis period[7,17]. We therefore, evaluated the extent of tissue losses by diabetes status and duration. Since new-onset diabetes is considered a consequence rather than a risk factor for pancreatic cancer[7,8,18,19], we categorized cases as no diabetes, recent-onset diabetes (≤4 years), and long-term diabetes (>4 years). Loss of muscle and adipose tissues was more pronounced among those with recent-onset diabetes compared to those with long-term or no diabetes (all *P* ≤ 0.0003) (Fig. 4). We next considered tissue loss separately for men and women, given known differences in body composition by sex[20,21]. Although present in both men and women, tissue loss was greater in men compared to women (all *P* ≤ 0.01, Fig. 4).

We next examined correlations between loss of weight and individual tissues in the prediagnosis period (Supplementary Fig. 12). We observed relatively strong correlations between loss of weight, visceral adipose tissue (Spearman correlation coefficient [$r_s$] = 0.61, *P* < 0.0001), and subcutaneous adipose tissue ($r_s$ = 0.62, *P* < 0.0001), but less strong correlation with skeletal muscle loss ($r_s$ = 0.43, *P* < 0.0001). Furthermore, loss of adipose tissue and skeletal muscle were not strongly correlated, suggesting that skeletal muscle loss may occur independently from weight and adipose tissue loss in some patients.

## Discussion

Early detection of pancreatic cancer remains difficult, due to few specific warning signs and early propensity for dissemination. However, pancreatic cancer-induced alterations in systemic metabolism may provide opportunities for earlier cancer detection[3,22]. In the current study, we examined how weight and peripheral tissues changed in the period leading up to a diagnosis of pancreatic cancer, using automated machine learning algorithms to interrogate prediagnosis CT imaging from pancreatic cancer cases and matched controls and a > 12,000 patient age-, sex-, and race-standardized CT reference set. We identified adipose tissue wasting primarily in the population with CT imaging 6 months prior to cancer diagnosis. In contrast, loss of skeletal muscle appeared to occur earlier, with a steady decline in skeletal muscle up to the time of cancer diagnosis. Notably, these changes were identified in equal magnitude among patients diagnosed with localized and metastatic pancreatic cancer, suggesting that cancer-induced metabolic changes may occur when pancreatic cancers can be treated with curative intent.

In previous studies using mouse models of pancreatic cancer, we observed peripheral tissue wasting early in disease progression[10]. This wasting occurred before the tumor spread beyond the pancreas and before measurable weight loss. Nevertheless, it is unclear if and when tissue wasting occurs in the early stages of pancreatic cancer in humans. A study of 68 pancreatic cancer patients with serial CT scans in the prediagnosis period provided initial data supporting tissue wasting prior to cancer diagnosis[13]. In that prior study evaluating raw tissue areas on CT imaging, subcutaneous adipose tissue appeared to decline first, followed by loss of skeletal muscle and visceral adipose tissue[13]. In contrast, in our study population evaluated with age-, sex-, and race-standardized tissue measurements, skeletal muscle appeared to decline earlier than adipose tissue in the time prior to cancer

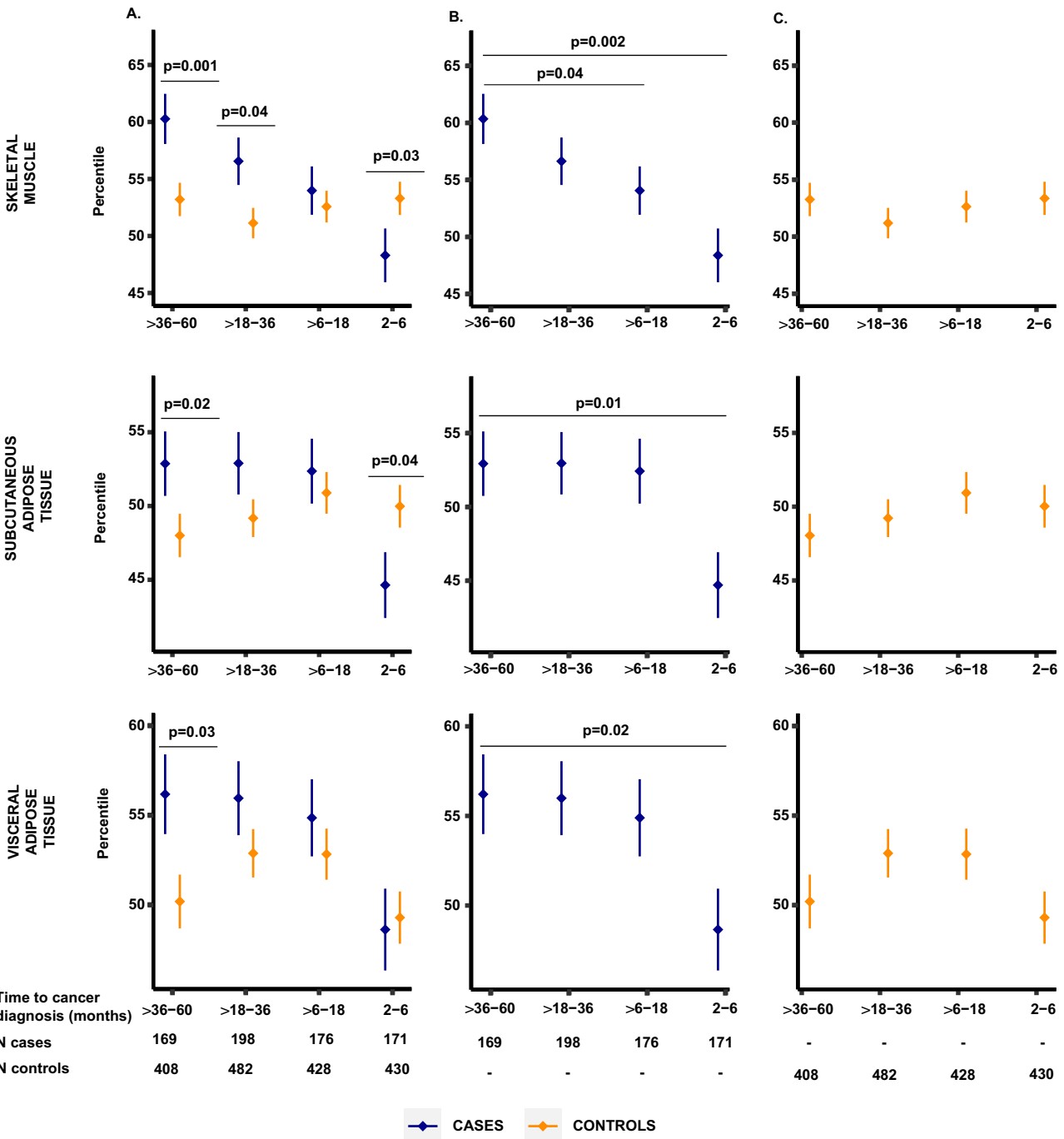

**Fig. 2 | Comparison of age-, sex-, and race-standardized tissue percentiles between cases and matched controls by time before pancreatic cancer diagnosis.** Data presented as mean values (dots) ± standard errors (bars). **A** Tissue percentiles in cases and matched controls by time before pancreatic cancer diagnosis. Horizontal lines mark time intervals with measurements significantly different between cases (blue) and controls (yellow) at a *P* value < 0.05, two-sided Wilcoxon rank-sum test. **B**, **C** Tissue percentiles in cases and controls, respectively, by time before pancreatic cancer diagnosis. Horizontal lines mark time intervals with measurements significantly different from the >36 to 60 months interval at a *P* value < 0.05, Wilcoxon rank-sum test. Source data are provided as a Source Data file.

diagnosis. This difference could be due to the evaluation of raw tissue areas without age-, sex-, and race-standardization and a smaller patient population with serial prediagnosis CT scans in the aforementioned study. Nevertheless, the results of both studies suggest that tissue changes occur in the 18 months prior to pancreatic cancer diagnosis.

To examine tissue changes, we chose a reference time period of >36 to 60 months before pancreatic cancer diagnosis, assuming that tissue loss would not have begun yet at this time. However, for skeletal muscle, the populations in each time period prior to diagnosis demonstrated lower skeletal muscle areas than those at the >36 to

60 months prediagnosis period, such that we cannot rule out that subtle skeletal muscle wasting could occur even prior to 36 months. Skeletal muscle measurements were statistically significantly lower compared to the reference period starting at >6 to 18 months before diagnosis, and steady loss of skeletal muscle appeared to occur up to the time of diagnosis.

Based on the tissue measurements in the current study, it is likely that changes in tissue areas over time rather than measured at a single time point will be most informative in assessing risk for pancreatic cancer. At >36 to 60 months before pancreatic cancer diagnosis, cases

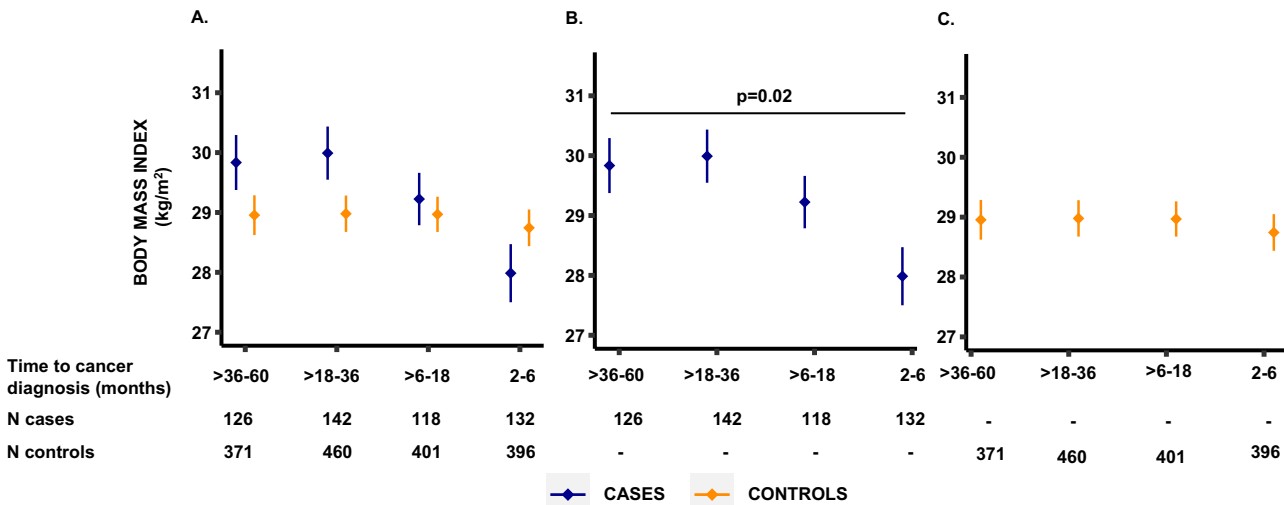

Fig. 3 | **Comparison of body mass index between cases and matched controls by time before pancreatic cancer diagnosis.** Data presented as mean values (dots) ± standard errors (bars). **A** BMI in cases and matched controls by time before pancreatic cancer diagnosis. Horizontal lines mark time intervals with measurements significantly different between cases (blue) and controls (yellow) at a *P* value <0.05, two-sided Wilcoxon rank-sum test. **B, C** BMI in cases and controls, respectively, by time before pancreatic cancer diagnosis. Horizontal lines mark time intervals with measurements significantly different from the >36 to 60 months interval at a *P* value <0.05, Wilcoxon rank-sum test. Source data are provided as a Source Data file.

**Table 2 | Change in weight and tissue measurements between prediagnosis and at-diagnosis CT scans in pancreatic cancer cases**

|  | Prediagnosis scan[a] | At-diagnosis scan[a] | Absolute change between scans[a] | % change between scans[a] | % change between scans per year of follow-up | P value[b] |
|---|---|---|---|---|---|---|
| Weight, kg | 83.0 (27.1) | 76.9 (22.8) | −5.2 (8.4) | −6.3 (9.6) | −4.0 (9.3) | <0.0001 |
| Body mass index, kg/m² | 28.5 (7.5) | 26.5 (7.1) | −1.8 (2.8) | −6.3 (9.6) | −4.1 (9.3) | <0.0001 |
| Visceral AT area, cm² | 179.0 (188.6) | 150.3 (165.6) | −20.3 (56.0) | −11.5 (33.5) | −7.1 (27.8) | <0.0001 |
| Subcutaneous AT area, cm² | 210.1 (158.1) | 187.0 (137.9) | −20.9 (55.4) | −11.4 (25.0) | −6.1 (20.3) | <0.0001 |
| Skeletal muscle area, cm² | 137.7 (53.1) | 128.3 (47.5) | −8.6 (16.1) | −5.9 (11.1) | −3.5 (7.0) | <0.0001 |

*AT* adipose tissue, *CT* computed tomography.
[a]Median (interquartile range).
[b]Two-sided Wilcoxon signed-rank test.

had greater adipose tissue and skeletal muscle compared to controls, likely reflecting that overweight and obesity are risk factors for pancreatic cancer[23]. However, at time points closer to the diagnosis of pancreatic cancer, peripheral tissue areas were reduced in cases, ultimately becoming lower in cases compared to controls. Skeletal muscle changes might therefore serve as a longitudinal surveillance marker, with potential utility among individuals at elevated pancreatic cancer risk, such as those with familial or genetic risk, pancreatic cystic lesions, or adult new-onset diabetes. Adult new-onset diabetes is particularly notable in this regard, as these individuals had greater tissue wasting in the period prior to their pancreatic cancer diagnosis[7,17]. Given the potential for tissue changes to occur with other conditions, general population screening for pancreatic cancer would need to consider multiple factors in combination, and solely evaluating adipose tissue and skeletal muscle is unlikely to be a fruitful strategy. Thus, future studies will need to evaluate high-risk populations, define thresholds of skeletal muscle change over time with appropriate sensitivity and specificity for predicting pancreatic cancer diagnosis, and account for the need to standardize tissue measurements by age, sex, and race. Furthermore, while skeletal muscle was measured using CT imaging in the current study, future studies should evaluate the potential of other less invasive and expensive methods, such as dual-energy X-ray absorptiometry[24] or blood-based biomarkers of muscle wasting[25].

Strengths of this study include a large number of cases and matched healthy controls with available CT imaging in the period preceding diagnosis of pancreatic cancer. These individuals originated from two large, geographically distinct health systems, and tissue areas were standardized to a large reference set of more than 12,000 outpatient CT scans to minimize the variability of raw tissue measurements in relation to age, sex, and race. Standardization of tissue areas is of particular importance given these areas vary with age, sex, and race, as we have shown previously[26], and tend to decline over time in older adults. Automated machine learning algorithms also allowed large numbers of CT scans to be analyzed with high reproducibility for cases, controls, and the large outpatient imaging reference set. The current study has several limitations to note. We analyzed CT imaging obtained up to >36–60 months before diagnosis, and it is possible that changes in some tissues could start at an even earlier time point. It is possible that cases with prediagnosis imaging obtained multiple years before diagnosis have different disease biology from cases with scans performed in the several months prior to diagnosis or with no prior CT imaging. Furthermore, tissue measurements in pancreatic cancer cases by time before diagnosis were compared between different groups of cases. While these measurements were age-, sex, and race-standardized, cases could potentially differ by other variables that influenced tissue areas. Therefore, longitudinal assessment of tissue areas among patients at elevated risk for pancreatic cancer will be important to definitively

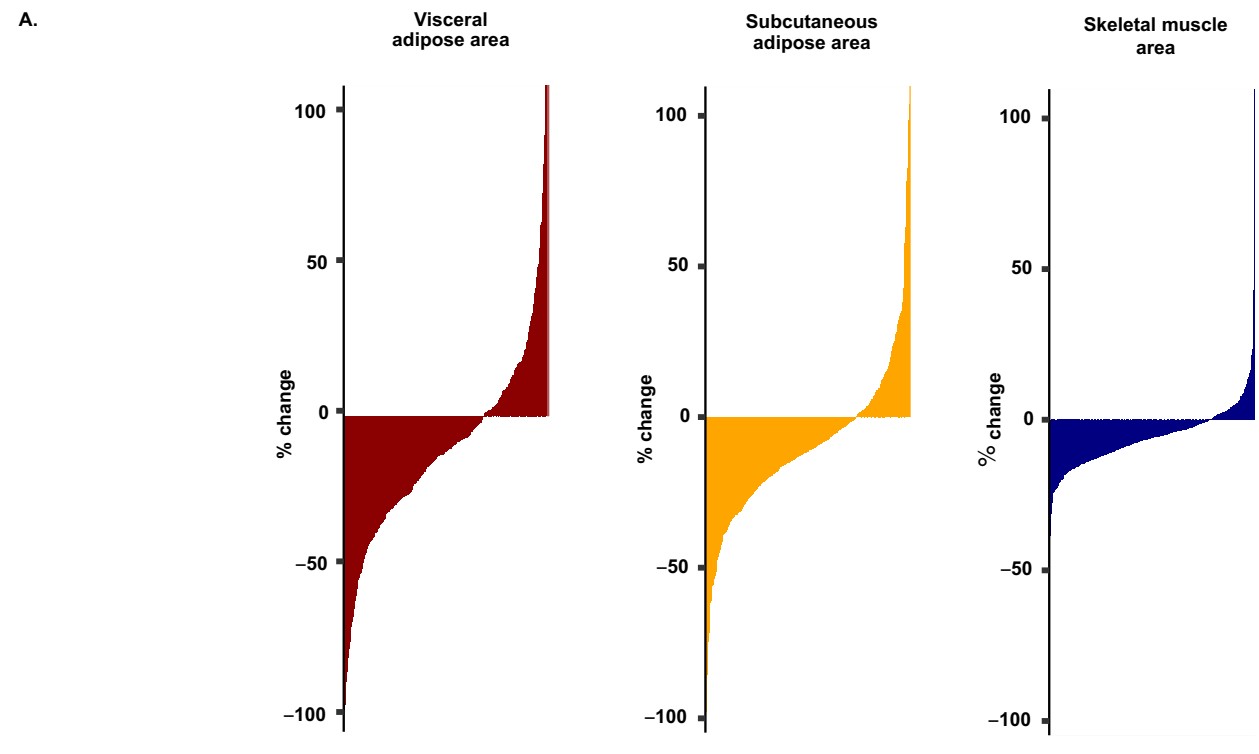

**Fig. 4 | Change in adipose and skeletal muscle tissue measurements in prediagnostic period in individual patients. A** Waterfall plots of visceral (red), subcutaneous (yellow) and skeletal muscle tissue (blue) measurement changes between prediagnosis and at-diagnosis CT scan. Each vertical line corresponds to one pancreatic cancer case. **B** Change in adipose and skeletal muscle tissue between prediagnosis and at-diagnosis CT scan by select patient characteristics. Diabetes status and duration related to the time of pancreatic cancer diagnosis. P-values calculated using two-sided Wilcoxon rank sum test. Source data are provided as a Source Data file.

characterize intra-patient changes over time. We did not match cases and controls on BMI or other measures of body shape. However, this allowed us to evaluate changes in BMI prior to cancer diagnosis in comparison with skeletal muscle and adipose tissue measurements by CT imaging. It is possible that prediagnosis CT imaging might have been performed for symptoms or signs related to the subsequently diagnosed cancer. Nevertheless, the indications for imaging were largely similar between cases and controls, and the imaging did not identify a clear pancreas mass based on a standard of care evaluation. Lastly, this analysis did not include individuals of racial or ethnic groups for whom reference curves for tissue measurements have not been developed, such that additional work will be required in these populations.

In conclusion, the current study identified tissue wasting on routine CT imaging in the years preceding the diagnosis of pancreatic cancer, including among patients who were diagnosed with localized, potentially curable disease. Future studies should evaluate the utility of longitudinal tissue measurements for earlier detection of pancreatic cancer among at-risk groups.

## Methods
### Ethics statement
This study complies with all relevant ethical regulations. Institutional review boards at Dana–Farber Cancer Institute/Mass General Brigham (DFCI/MGB) and Kaiser Permanente Northern California (KPNC)

approved the study and provided a waiver for informed consent. Informed consent was waived as IRBs at both institutions determined the risk to patients was minimal relative to the knowledge that might be gained.

## Study populations

To study tissue wasting in the time period preceding pancreatic cancer diagnosis, we identified pancreatic cancer cases diagnosed between 2003 and 2017 within the DFCI/MGB and KPNC care networks. MGB is an integrated healthcare system founded by Massachusetts General Hospital and Brigham and Women's Hospital, including community and specialty hospitals, a physician network, community health centers, and other health-related services. DFCI is a comprehensive cancer treatment and research institution with affiliation and shared electronic medical record (EMR) with MGB. KPNC is an integrated healthcare delivery organization that combines an insurance provider, a physician-owned practice group, 21 hospitals, and over 200 outpatient clinics. KPNC cares for over 4 million members encompassing the San Francisco Bay Area and the Central Valley from Sacramento in the north to Fresno in the south. Within DFCI/MGB, pancreatic cancer cases were identified from a DFCI institutional databank and from an electronic database via ICD 9/10 code [ICD9: 157.0 – 157.9; ICD10: C25.0 – C25.9] in the MGB system with manual verification of diagnosis. Within KPNC, cases were identified from a registry of all cancer cases diagnosed within the integrated health system via the same ICD codes, and histologic description was subsequently manually confirmed. The cancer diagnosis was defined as the date of histologic confirmation of pancreatic adenocarcinoma, including variants such as poorly differentiated carcinoma and adenosquamous carcinoma. Patients with pancreatic neuroendocrine tumors or other rare pancreatic cancer variants were excluded.

Within DFCI/MGB and KPNC, we identified 987 pancreatic cancer cases with one or more abdominal computed tomography (CT) scans in the period of 2 months to 5 years prior to their cancer diagnosis. For those cases with two or more prediagnosis scans in this time window, one scan was chosen at random as the reference prediagnosis scan for the primary analyses. Cases were excluded if they had a history of cancer in 5 years prior to the prediagnosis scan, unknown date of prior cancer, or scan indication related to pancreatic cancer diagnosis. We also excluded cases belonging to demographic groups for which age- and race-standardized reference curves for tissue measurements were not available[26]. To provide a reference population for comparison to the pancreatic cancer cases, we matched control participants from DFCI/MGB and KPNC with available abdominal CT imaging and no history of cancer except nonmelanoma skin cancers within 5 years prior and 6 months after the selected scan. Pancreatic cancer cases were matched with 1 to 3 controls by age (±2 years), sex, race (white, black, other), care network (DFCI/MGB, KPNC), year of CT scan (±3 years), and use of intravenous contrast with CT imaging (yes, no). The total case-control study population consisted of 714 cases with prediagnosis CT scans and 1748 matched controls (Fig. 1). Among the 714 cases with prediagnosis CT imaging, 529 cases also had available at-diagnosis CT imaging within 60 days of pathological diagnosis for comparative analyses (Fig. 1).

## Study variables

For pancreatic cancer cases and matched controls, information at the time of the prediagnosis scan was collected for age, self-reported sex, race/ethnicity, height, weight, smoking status, alcohol use, history and duration of diabetes, personal cancer history, and indication for CT imaging. For cases, we also collected information at the time of diagnosis, including age, weight, history, duration of diabetes, disease stage, tumor location in the pancreas, CA19-9 level, and dates of histologic diagnosis, death, and last follow-up. For DFCI/MGB participants, all variables were extracted from the EMR. For KPNC, all variables were extracted from the tumor registry and the Virtual Data

Warehouse (VDW), a set of standardized data developed as a collaboration by the Health Care Systems Research Network to facilitate public domain health and health services research. In KPNC, the VDW includes demographic, socioeconomic, diagnostic, treatment, and outcomes data for more than 12 million patients seen since 1996.

## Tissue compartment analysis

To quantify skeletal muscle, visceral adipose tissue, and subcutaneous adipose tissue among case and control participants, we used an automated deep-learning pipeline for tissue segmentation from abdominal CT scans at the L3 vertebral body, as previously described and validated[26]. Tissue compartment areas measured at the L3 vertebral body have been shown to correlate highly with the total body volume of skeletal muscle and adipose tissue[27,28]. In the DFCI/MGB and KPNC case-control population, all CT scans were assessed in a deidentified, blinded manner with random distribution. Quality control processes rejected scans that did not include the L3 level or included incomplete reconstructions of the L3 anatomy.

We and others have previously demonstrated that skeletal muscle and adipose tissue areas differ by patient age, sex, and race[20,21,26,29]. In a previously published study, we generated standardized tissue reference curves from abdominal CT scans of > 12,000 individuals without a prior history of cancer to account for these demographic differences and facilitate normalized comparisons within diverse populations[26]. Reference curves were developed for white non-Hispanic females and males (18–90 years), black females (18–75 years), and black male (18–70 years) participants, but could not be developed for other racial and age groups due to insufficient participant numbers. Using these reference curves, each tissue measurement can be assigned a z-score value for their age, sex, and race, which corresponds to that measurement's distance from the population-specific median. Furthermore, these z-scores allow each measurement to be related to others of the same age, sex, and race using a population percentile, similar to how children are followed on growth curves during development[30]. The z-scores adjust for demographic-specific offsets in tissue compartments, including nonlinear effects of age[26]. Thus, we converted skeletal muscle and adipose tissue areas from CT scans into z-scores and percentiles for all pancreatic cancer cases and matched controls.

## Statistical analysis

To compare the change in tissue area measurements with the approaching diagnosis of pancreatic cancer, we divided cases and matched controls into groups based on time elapsed between prediagnosis CT imaging and pancreatic cancer diagnosis (2 to 6 months, >6 to 18 months, >18 to 36 months, and >36 to 60 months), similar to a previous study of prediagnosis changes in metabolism[13]. At each time point, tissue measurements between cases and controls were compared using the Wilcoxon rank-sum test. To evaluate change in tissue measurements over time, we compared measurements at each interval to the earliest available time interval (>36–60 months) using the Wilcoxon rank-sum test, for cases and control separately.

Among pancreatic cancer cases with available prediagnosis and at-diagnosis CT scans, tissue measurements at two time points were compared using the Wilcoxon signed-rank test. We used waterfall plots to visualize the percent change in tissue compartments in individual cases. For scaling purposes, these plots did not include cases with more than a 100% increase in skeletal muscle ($N = 2$), visceral adipose ($N = 12$), or subcutaneous adipose tissue ($N = 5$). Stratified analyses were performed to compare changes between prediagnosis and at-diagnosis scans by disease stage, tumor location, history and duration of diabetes, and sex. Spearman correlation coefficients were used to quantify the correlation between changes in weight and tissue compartments.

All P values were two-sided. Statistical analyses were performed with SAS 9.1 (SAS Institute, Cary, North Carolina). Plots were generated using RStudio 2022.07.1 Build 554.

**Reporting summary**

Further information on research design is available in the Nature Portfolio Reporting Summary linked to this article.

## Data availability

Imaging studies used in this research include identifiers (names, dates of birth, medical record numbers) that are considered protected health information under local laws and are directly stored in the pixel data by the originating scanners. Thus, raw clinical imaging data cannot be provided. The DFCI/MGB and KPNC IRBs waived the requirement for informed consent and approved the use of the study data by the research team and limited use solely to accomplish the study aims. Deidentified data for DFCI/MGB can be obtained from the corresponding author (B.M.W.) with appropriate IRB approval and interinstitutional data use agreement. For KPNC data, access can be obtained by contacting the KPNC research team (S.K.V.D.E. or B.J.C.) for review and application for IRB approval. All processed data used to generate figures has been provided in the Source Data file. Source data are provided with this paper.

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

## Acknowledgements

A.B. is supported by the NIH K07 CA222159 grant and Bob Parsons Memorial Fellowship. N.K. is supported by Veterans Affairs Health Services Research and Development Career Development Award (1 IK2 HX003346-01A2). M.B.Y. is supported by Hale Family Center for Pancreatic Cancer Research and Stand Up to Cancer (#SU2C-AACR-DT25-17). K.N. acknowledges support from the Broman Family Fund for Pancreatic Cancer Research. M.G.V.H. acknowledges support from the Lustgarten Foundation, the MIT Center for Precision Cancer Medicine, the Ludwig Center at MIT, and NIH grants R35 CA242379 and P30 CA1405141. B.M.W. is supported by the Hale Family Center for Pancreatic Cancer Research, Lustgarten Foundation Dedicated Laboratory program, NIH grant U01 CA210171, NIH grant P50 CA127003, Stand Up to Cancer, Pancreatic Cancer Action Network, Noble Effort Fund, Wexler Family Fund, Parsons Pancreatic Cancer Early Detection Fund, and Promises for Purple.

## Author contributions

A.B., M.H.R., T.K.S., S.K.V.D.E., B.J.C., and B.M.W. had full access to all the data in the study and took responsibility for the integrity of the data and the accuracy of the data analysis. Concept and design: A.B., M.H.R., T.K.S., N.K., L.K.B., C.Y., S.K.V.D.E., B.J.C., and B.M.W. Acquisition, analysis, or interpretation of data: A.B., M.H.R., T.K.S., N.K., V.L., L.K.B., M.L., L.C., S.D., A.G., J.H., K.S., C.Y., Q.W., A.A.L., S.K.V.D.E., B.J.C., and B.M.W.

Statistical analysis: A.B., M.H.R. Drafting of the manuscript: A.B., M.H.R., T.K.S., S.K.V.D.E., B.J.C., and B.M.W. Critical revision of the manuscript for important intellectual content: A.B., M.H.R., T.K.S., N.K., V.L., L.K.B., M.L., L.C., S.D., A.G., J.H., K.S., C.Y., Q.W., A.A.L., L.H.B., M.B.Y., K.N., J.A.N., A.J.A., S.N.B., M.G.V.H., S.K.V.D.E., B.J.C., and B.M.W. Obtained funding: A.B., M.H.R., T.K.S., N.K., S.K.V.D.E., B.J.C., and B.M.W. Administrative, technical, or material support: A.B., M.H.R., T.K.S., N.K., V.L., L.K.B., M.L., L.C., S.D., A.G., J.H., K.S., C.Y., Q.W., A.A.L., L.H.B., M.B.Y., K.N., J.A.N., A.J.A., S.N.B., M.G.V.H., S.K.V.D.E., B.J.C., and B.M.W. Study supervision: S.K.V.D.E., B.J.C., and B.M.W.

## Competing interests

M.G.V.H. is a scientific advisor for Agios Pharmaceuticals, iTeos Therapeutics, Sage Therapeutics, Faeth Therapeutics, Droia Ventures, and Auron Therapeutics on topics unrelated to the presented work. B.M.W. received research funding from Celgene, Eli Lilly, Novartis, and Revolution Medicine and consulting fees from Celgene, Grail, Mirati, and Third Rock Ventures unrelated to the presented work. M.B.Y. received research grant from Janssen Pharmaceuticals and fees for peer review services from UpToDate, unrelated to the presented work. The remaining authors declare no competing interests.

## Additional information

[1]Department of Medical Oncology, Dana-Farber Cancer Institute, Boston, MA, USA. [2]Department of Radiology, Brigham and Women's Hospital and Harvard Medical School, Boston, MA, USA. [3]Kaiser Permanente San Francisco, San Francisco, CA, USA. [4]Center for Innovations in Quality, Effectiveness, and Safety (IQuESt), Michael E. DeBakey Veterans Affairs Medical Center; Section of Gastroenterology and Hepatology, Department of Medicine, Baylor College of Medicine, Houston, TX, USA. [5]Division of Research, Kaiser Permanente Northern California, Oakland, CA, USA. [6]Department of Clinical Science, Intervention and Technology, Karolinska Institutet, Stockholm, Sweden. [7]Division of Gastroenterology, Hepatology and Endoscopy, Brigham and Women's Hospital and Harvard Medical School, Boston, MA, USA. [8]Department of Medicine, Brigham and Women's Hospital and Harvard Medical School, Boston, MA, USA. [9]Department of Pathology, Brigham and Women's Hospital and Harvard Medical School, Boston, MA, USA. [10]Harvard-MIT Division of Health Sciences and Technology, Institute for Medical Engineering and Science, Massachusetts Institute of Technology, Cambridge, MA, USA. [11]Koch Institute for Integrative Cancer Research, Massachusetts Institute of Technology, Cambridge, MA, USA. [12]Department of Electrical Engineering and Computer Science, Massachusetts Institute of Technology, Cambridge, MA, USA. [13]Broad Institute of Massachusetts Institute of Technology and Harvard, Cambridge, MA, USA. [14]Wyss Institute, Harvard University, Boston, MA, USA. [15]Howard Hughes Medical Institute, Cambridge, MA, USA. [16]Department of Biology, Massachusetts Institute of Technology, Cambridge, MA, USA. [17]These authors contributed equally: Ana Babic, Michael H. Rosenthal, Tilak K. Sundaresan. [18]These authors jointly supervised this work: Stephen K. Van Den Eeden, Bette J. Caan, Brian M. Wolpin. ✉e-mail: brian_wolpin@dfci.harvard.edu

