## [Peer Review File · Nature Communications]

Adipose tissue and skeletal muscle wasting precede clinical diagnosis of pancreatic cancerReviewers' comments:

Reviewer #1 (Remarks to the Author):

This original manuscript reports the image-based muscle and fat mass in individuals who will later develop pancreatic cancer and in non-cancer controls matched age, sex, and race. The manuscript is well-written, and concise, and confirms prior studies that have shown that tissue loss precedes the diagnosis of pancreatic cancer. Specific comments are enclosed below.

-Table 1 should include statistical differences between groups.

-Analysis of BMI should be included in the main figures.

-Limitations should include that subjects were not matched by BMI, muscle, or fat mass at pre-illness.

Reviewer #2 (Remarks to the Author):

NCOMMS-22-45711: Skeletal muscle wasting precedes clinical diagnosis of pancreatic cancer

Overview: The authors evaluated CT-based measures of skeletal muscle and adipose tissue in patients prior to the clinical diagnosis of pancreatic cancer. Loss of both tissue types was observed (also relative to controls) as patients moved closer to cancer diagnosis.

Strengths:

- Relatively large pancreatic cohort (and corresponding control group) with CT scans prior to cancer diagnosis (larger than ref 13)

- Automated machine learning algorithms to measure muscle and fat.

- Well written and easy to follow.

Weaknesses:

- Ref 13 (Sah et al. Gastro 2019) seems to have found the same basic CT-based fat & muscle trends prior to PDAC diagnosis, which diminishes the impact of the current findings to more corroborative.

- Unavoidable lack of systemic timing of CT scans given the retrospective "catch as catch can" approach (again, unavoidable and not a criticism per se but nonetheless lacks rigor)

Specific comments:

1. Adipose tissue losses were also observed (albeit not as early). Why not include this in the title (and main conclusion)?

2. Intro: "...80% of patients are diagnosed after the tumor has spread beyond the pancreas when curative treatments are rarely feasible." May want to change to "...when curative treatment attempts...", since the majority of these cases are unfortunately not ultimately cured.

3. Nice overview of "what is known" in the Intro

4. Methods: Good to limit to pancreatic adenocarcinoma (w/ exclusion of PET). Poorly differentiated carcinoma and adenosquamous carcinoma were included – but what about mucinous variants (eg, those arising in IPMN or MCN)? In my opinion, these should be excluded and restrict to PDAC.

5. One can only wonder if some of the CT scans prior to clinical diagnosis were performed for symptoms related to undiagnosed cancer. I realize it may be difficult to ascertain, but were CT scans w/in ~1 year of clinical diagnosis at least reviewed for signs of missed pancreatic cancer? I'm surprised this wasn't even discussed, since the muscle changes are presumably paraneoplastic in part?

6. I might have hypothesized that pancreatic head tumors might be associated with less wasting (not more), since they tend to present earlier due to biliary obstruction (jaundice), compared with body/tail cancers.

7. "At 3 to 5 years before cancer diagnosis, cases had greater adipose tissue and skeletal muscle areas compared to matched controls" – why do you think this was? Could it be related to an inherent weakness in the control group?

8. Table 2 is presented without any reference to time. Should annualized changes be considered to avoid an “apples & oranges” comparison?

9. The conclusion that “skeletal muscle wasting may signal an upcoming diagnosis of pancreatic cancer” may be true but the specificity/PPV in the population would be so low that it likely renders it moot. There are simply too many other causes of sarcopenia for this to have an impact.

Reviewer #3 (Remarks to the Author):

Very nice study. Strengths include the many prediagnostic scans, from two sites, the CT algorithm for quantifying the tissue compartments, and the data available for the case/control analysis. No significant weaknesses.

The authors should comment on the small % of muscle mass that occurs in some very old adults just to put into context that this % is much smaller and less common than the loss seen in prediagnostic PDAC.

The sentence below should be clarified to make it clear it is referring to the pilot study (Ref 13) cited in the prior sentence.

“In this study evaluating raw tissue areas on CT imaging...”

To something like...

In that study (Ref 13) evaluating raw tissue areas on CT imaging...

Responses to Reviewers' comments

Reviewer #1 (Remarks to the Author):

This original manuscript reports the image-based muscle and fat mass in individuals who will later develop pancreatic cancer and in non-cancer controls matched age, sex, and race. The manuscript is well-written, and concise, and confirms prior studies that have shown that tissue loss precedes the diagnosis of pancreatic cancer. Specific comments are enclosed below.

-Table 1 should include statistical differences between groups.

We thank the Reviewer for this comment. In the revised manuscript, we now include *P*-values in Table 1 comparing cases and matched controls. As expected, no statistically significant differences were identified for the six matching factors: age, sex, race, care network, year of CT scan, and use of intravenous contrast with CT imaging. As would also be expected, we did see statistically significant differences in several well-established risk factors for pancreatic cancer, including history of diabetes, cigarette smoking, and personal history of cancer.

-Analysis of BMI should be included in the main figures.

We thank the Reviewer for this comment. We are happy to include analyses related to prediagnosis body-mass index in a figure in the main manuscript. In the revised manuscript, we have now included the prior Supplementary Figures 1 and 2 evaluating prediagnosis body-mass index as Figure 3 in the main manuscript.

-Limitations should include that subjects were not matched by BMI, muscle, or fat mass at pre-illness.

We thank the Reviewer for this comment. Pancreatic cancer cases were matched with 1 to 3 controls by age (± 2 years), sex (male, female), race (white, black, other), care network (DFCI/MGB, KPNC), year of CT scan (± 3 years), and use of intravenous contrast with CT imaging (yes, no). As noted by the Reviewer, we did not also match on body-mass index or measures of muscle or fat mass. Since the primary exposures assessed in the current analyses were prediagnosis adipose tissue and skeletal muscle, we did not match on these factors, as we wished to assess the association of prediagnosis tissue areas with pancreatic cancer development. Similarly, we did not match on prediagnosis body-mass index, as this would not allow us to assess the association of prediagnosis BMI with subsequent pancreatic cancer diagnosis. Based on the Reviewer's suggestion, we have included the lack of matching on BMI or other measures of muscle or fat mass in the limitations section of the Discussion within the revised manuscript.

On page 15 (paragraph 2) of the revised manuscript, we now state:

"We did not match cases and controls on BMI or other measures of body shape. However, this allowed us to evaluate changes in BMI prior to cancer diagnosis in comparison with skeletal muscle and adipose tissue measurements by CT imaging."

Reviewer #2 (Remarks to the Author):

NCOMMS-22-45711: Skeletal muscle wasting precedes clinical diagnosis of pancreatic cancer

Overview: The authors evaluated CT-based measures of skeletal muscle and adipose tissue in patients prior to the clinical diagnosis of pancreatic cancer. Loss of both tissue types was observed (also relative to controls) as patients moved closer to cancer diagnosis.

Strengths:

- *Relatively large pancreatic cohort (and corresponding control group) with CT scans prior to cancer diagnosis (larger than ref 13)*
- *Automated machine learning algorithms to measure muscle and fat.*
- *Well written and easy to follow.*

We thank the Reviewer for these comments. As noted by the Reviewer, this large study was made possible by a collaboration between two health care networks and the design and utilization of a machine learning algorithm to facilitate high-throughput, hands-off tissue quantification from thousands of CT images. This multi-year effort led to the largest study of prediagnosis imaging conducted in pancreatic cancer and highlights previously unappreciated biology related to tissue wasting prior to cancer diagnosis. We appreciate the Reviewer's comment that the manuscript was well written and easy to follow.

Weaknesses:

- *Ref 13 (Sah et al. Gastro 2019) seems to have found the same basic CT-based fat & muscle trends prior to PDAC diagnosis, which diminishes the impact of the current findings to more corroborative.*
- *Unavoidable lack of systemic timing of CT scans given the retrospective "catch as catch can" approach (again, unavoidable and not a criticism per se but nonetheless lacks rigor)*

We appreciate the Reviewer's comments. We would like to emphasize that our study utilized several methodological and conceptual innovations that have advanced our understanding of tissue changes in the prediagnostic period, with important implications for early diagnosis of pancreatic cancer.

Tissue wasting, commonly described as cancer-associated cachexia, is one of the hallmarks of advanced pancreatic cancer and other advanced malignancies. However, nearly all studies have focused on characterizing the biology of cachexia late in the disease process. Importantly, it is unknown when tissue wasting develops during cancer development. Furthermore, the potential utility of markers of tissue wasting remains unexplored as part of cancer early detection programs.

Within our own work, we have steadily pursued this area over the past 10 years, with this being the third manuscript in stepwise fashion to elucidate underlying biology of tissue wasting in early pancreatic cancer using both mouse models and patient data. The first two manuscripts were published previously: Mayers et al. *Nature Medicine*, 2014, PMID: 25261994; and Danai et al. *Nature*, 2018, PMID: 29925948. In the first manuscript (Mayers et al.), we used plasma samples collected within large prospective cohorts to evaluate metabolic changes prior to a pancreatic cancer diagnosis. This work led us to a set of characteristic changes in prediagnosis blood samples collected several years prior to a pancreatic cancer diagnosis in patients that we then validated in genetically engineered mouse models of pancreatic cancer. Furthermore, the metabolites that were elevated in the blood of humans and mice prior to diagnosis of pancreatic cancer were found to be liberated from peripheral tissues, particularly adipose tissue and skeletal muscle. This work established that metabolic changes are occurring before diagnosis in patients with pancreatic cancer and that these metabolites are originating from peripheral tissues in genetically engineered mouse models. In a follow-up manuscript (Danai et al.), we demonstrated that

both adipose tissue and skeletal muscle wasting occurs early in the development of pancreatic cancer in mouse models and that tumor growth within the pancreas is necessary for the development of this wasting phenotype in mice. Furthermore, decreased exocrine pancreatic function was partly a driver of the wasting phenotype in mice.

Given our prior findings suggesting metabolic changes prior to pancreatic cancer diagnosis, we undertook the current study (Babic et al.), which is the largest effort pursued to collect prediagnosis imaging studies in patients who develop pancreatic cancer. This study required: (1) extensive collaboration between investigators at two large health care systems to acquire large numbers of prediagnosis computed tomography (CT) scans, (2) development of machine learning algorithms for automated skeletal muscle and adipose tissue measurements from CT imaging to rapidly and accurately interrogate thousands of CT scans, (3) generation of a standardized reference set of >10,000 individuals with tissue measurements to account for substantial differences in skeletal muscle and adipose tissue by age, sex, and race, (4) rigorous matching of multiple controls to each case accounting for age, sex, race, health system, year of CT scan, and use of IV contrast, to ensure our ability to determine whether tissue changes were distinct to patients who developed pancreatic cancer or were also seen due to changes in population characteristics over time, and (5) collection of comprehensive clinical data from patients, including stage of cancer and circulating levels of CA19-9 at diagnosis, to evaluate whether identified changes were seen with disease localized to the pancreas or only once metastases had developed.

After multiple years of work to accomplish the above required elements for the current study, we were able to make multiple novel observations in patients within the current manuscript, which are particularly relevant as we consider novel approaches to earlier detection of pancreatic cancer:

(1) We showed that using age-, sex-, and race-standardized tissue percentiles, cases with imaging >36 to 60 months before diagnosis had higher percentiles of skeletal muscle, visceral adipose tissue, and subcutaneous adipose tissue than matched controls, indicating the importance of body composition as a risk factor for pancreatic cancer development. Although body-mass index (BMI) is a known risk factor for pancreatic cancer, no prior studies have evaluated whether excess of specific tissue compartments might predict future risk of pancreatic cancer.

(2) We showed that with the approaching pancreatic cancer diagnosis, relationships inverted for adipose tissue and skeletal muscle, with percentiles for these tissue compartments being lower in cases compared to controls as the diagnosis approached. Patterns of tissue composition change in cases compared to controls have not been examined in prior studies, and the identified pattern of tissue compartment change in the time before a pancreatic cancer diagnosis has not been shown previously.

(3) We demonstrated that BMI did not capture the tissue wasting phenotype as accurately as direct tissue measurements, highlighting the need to move beyond rough measures of body composition (such as BMI) when considering phenotypes such as tissue wasting.

(4) We demonstrated that the control population did not experience the same tissue changes as did the cases, highlighting the specificity of the observed phenotype to those who subsequently develop pancreatic cancer.

(5) We demonstrated that the large majority of pancreatic cancer cases experienced tissue loss in the time prior to their cancer diagnosis, with >70% of patients losing skeletal muscle and adipose tissue. The

prevalence of this wasting phenotype has not been examined in any prior studies, and the high rate of tissue losses is important information as we consider this biology for utility in early detection programs.

(6) We demonstrated that the observed tissue losses were not stage dependent, with equal numbers of patients developing skeletal muscle and adipose tissue wasting prior to diagnosis with non-metastatic and metastatic disease. This is a critical finding, as it suggests that the tissue wasting phenotype may have utility in early diagnosis of localized pancreatic cancer. It also verified what we observed in mouse models, where tissue wasting was seen with early cancers confined to the pancreas.

(7) We demonstrated that the degree of tissue wasting was not correlated with circulating levels of CA19-9, which is a well-known marker of disease burden in patients with pancreatic cancer. This again highlights that tissue wasting can be identified in patients with early-stage disease, where curative therapies are considered.

(8) We demonstrated that tissue wasting was most pronounced among pancreatic cancer patients who developed new-onset diabetes (NOD), i.e. diabetes in the several years prior to a pancreatic cancer diagnosis. Among individuals with NOD diagnosed after the age of 50 years, 0.4% to 0.8% will be diagnosed with pancreatic cancer in the next 3-4 years. However, it has been difficult to leverage this relationship for early detection of pancreatic cancer given that diabetes is common in the general population and most patients with NOD will not develop pancreatic cancer. However, additional features that can further refine risk estimates in this population would be extremely useful for early detection efforts. Thus, our novel finding that substantial tissue wasting accompanies NOD may have utility as screening approaches are considered for this population (e.g., Maitra et al. *Pancreas*, 2018; PMID: 30325864; Wu et al. *Clin Transl Gastro*, 2023, PMID: 35470312)

We are aware of one other published manuscript that has considered CT-measured tissue compartments before a diagnosis of pancreatic cancer (Sah et al. *Gastroenterology*, 2019, PMID: 30677401). However, this important study was primarily focused on other aspects of pancreatic cancer biology, evaluating changes in blood glucose, lipids, weight and temperature, with only a small component considering skeletal muscle and adipose tissue measurements. Notably, this study was a case-only evaluation that had a limited sample size with prediagnosis CT imaging, did not include control groups, had limited power to define timing of when tissue wasting occurs prior to pancreatic cancer diagnosis, and did not examine how tissue wasting develops in relation to disease burden. Furthermore, this study quantified raw measurements of muscle and adipose tissue without considering the differences in these tissues by age, race, and sex. As noted above, our large and rigorously conducted study provided multiple novel findings beyond those described in the important study by Sah et al., and notably highlighted the potential for tissue wasting to be used in early detection of this aggressive and highly lethal malignancy.

Regarding the issue of CT timing raised by the Reviewer, we agree with the Reviewer that CT scans were performed at varying time intervals prior to pancreatic cancer diagnosis. As the Reviewer notes, this is unavoidable, as individuals in the general population do not undergo regular interval CT imaging and the development of pancreatic cancer is not known in advance. However, our large sample size ensured that we had substantial numbers of pancreatic cancer cases and matched controls in each of four prediagnosis time periods, such that stable and externally standardized estimates of tissue areas could be provided.

Specific comments:

1. *Adipose tissue losses were also observed (albeit not as early). Why not include this in the title (and main conclusion)?*

We thank the reviewer for this comment. We agree that adipose tissue changes were noted closer to the time of pancreatic cancer diagnosis and could also be highlighted in the title and abstract. Based on the Reviewer's suggestion, we have now made the following changes within the revised manuscript. The title of the revised manuscript and the main conclusion now also include the observed loss of adipose tissue.

Title:

"Adipose tissue and skeletal muscle wasting precedes clinical diagnosis of pancreatic cancer"

Abstract:

"Adipose tissue loss was identified up to 6 months, and skeletal muscle wasting was identified up to 18 months before the clinical diagnosis of pancreatic cancer and was not present in the matched control population."

2. *Intro: "...80% of patients are diagnosed after the tumor has spread beyond the pancreas when curative treatments are rarely feasible." May want to change to "...when curative treatment attempts...", since the majority of these cases are unfortunately not ultimately cured.*

As suggested by the Reviewer, the text has now been edited on page 4 (paragraph 1) of the revised manuscript:

"Due to lack of specific early symptoms and no clinically available biomarkers for early disease detection, more than 80% of patients are diagnosed after the tumor has spread beyond the pancreas when curative treatments attempts are rarely feasible."

3. *Nice overview of "what is known" in the Intro*

We thank the Reviewer for this comment.

4. *Methods: Good to limit to pancreatic adenocarcinoma (w/ exclusion of PET). Poorly differentiated carcinoma and adenosquamous carcinoma were included – but what about mucinous variants (eg, those arising in IPMN or MCN)? In my opinion, these should be excluded and restrict to PDAC.*

We thank the Reviewer for this comment. As noted by the Reviewer, we included all cases with pancreatic adenocarcinoma or poorly differentiated carcinoma, including those with a component of squamous differentiation. We do not have systematically collected data on whether these tumors had a mucinous component or whether they arose from an intraductal papillary mucinous neoplasm (IPMN) or mucinous cystic neoplasm (MCN). However, cases were only included if they had invasive carcinoma, so patients with non-invasive cystic lesions with or without high-grade dysplasia would not be included in our analyses. Furthermore, patients undergoing regular follow-up for pancreatic cysts were excluded from analyses, since these patients were being followed for an abnormality in the pancreas.

5. *One can only wonder if some of the CT scans prior to clinical diagnosis were performed for symptoms*

related to undiagnosed cancer. I realize it may be difficult to ascertain, but were CT scans w/in ~1 year of clinical diagnosis at least reviewed for signs of missed pancreatic cancer? I'm surprised this wasn't even discussed, since the muscle changes are presumably paraneoplastic in part?

We agree with the Reviewer that it is possible some of the CT scans in the time window closest to diagnosis might have been performed due to early symptoms of the cancer, such as gastrointestinal symptoms or pain. Due to the large number of CT scans included in this analysis, we did not perform a manual review of the images in cases and controls to evaluate for presence of suspicious lesions in the pancreas. However, we know that these images did not clearly indicate a diagnosis of pancreatic cancer, as we reviewed the clinical imaging reports and excluded patients where a pancreas mass was identified. In the future, it would be interesting to consider whether skeletal muscle and adipose tissue wasting might provide additional information supporting a pancreatic cancer diagnosis when imaging is equivocal or unclear related to a mass in the pancreas. Based on the Reviewer's comment, we have now included the potential for imaging to have been performed for symptoms or signs of the undiagnosed cancer in the Discussion section of the revised manuscript.

On page 15 (paragraph 2) of the revised manuscript within the Discussion section, we now state:

"It is possible that prediagnosis CT imaging might have been performed for symptoms or signs related to the subsequently diagnosed cancer. Nevertheless, the indications for imaging were largely similar between cases and controls, and the imaging did not identify a clear pancreas mass based on standard of care evaluation."

6. I might have hypothesized that pancreatic head tumors might be associated with less wasting (not more), since they tend to present earlier due to biliary obstruction (jaundice), compared with body/tail cancers.

We agree with the Reviewer that tumors arising in the head of pancreas are more likely to be detected at an earlier stage due to biliary obstruction. However, these tumors are also more likely to lead to increased disruption of pancreatic exocrine function, which we have previously shown to be associated with peripheral tissue loss in mouse models of pancreatic cancer. In the current study, no statistically significant differences in adipose tissue or skeletal muscle wasting were identified by site of the primary tumor.

7. "At 3 to 5 years before cancer diagnosis, cases had greater adipose tissue and skeletal muscle areas compared to matched controls" – why do you think this was? Could it be related to an inherent weakness in the control group?

We thank the Reviewer for this comment. We think that higher adipose tissue and muscle skeletal areas at this timepoint likely reflect overweight and obesity as risk factors for pancreatic cancer. Although this has not been investigated in prior studies, one might predict that individuals who go on to develop pancreatic cancer would have greater adipose tissue and/or skeletal muscle areas before the onset of cancer-induced tissue wasting. We believe that the finding of increased tissue areas 3 to 5 years before diagnosis supports this hypothesis, and therefore is consistent with the epidemiology of this cancer type rather than a weakness of the control group. It will be interesting in the future to better characterize how different tissues contribute to overweight and obesity as risk factors for pancreatic cancer.

8. Table 2 is presented without any reference to time. Should annualized changes be considered to avoid an “apples & oranges” comparison?

We appreciate this comment from the Reviewer. In response to the Reviewer’s suggestion, we have now added annualized percent change to Table 2 in the revised manuscript.

9. The conclusion that “skeletal muscle wasting may signal an upcoming diagnosis of pancreatic cancer” may be true but the specificity/PPV in the population would be so low that it likely renders it moot. There are simply too many other causes of sarcopenia for this to have an impact.

We agree with the Reviewer that decline in skeletal muscle area by CT imaging is unlikely to be a stand-alone biomarker for an impending pancreatic cancer in the general population, since skeletal muscle wasting can be due to a variety of conditions and pancreatic cancer is a relatively rare disease. However, we do believe that this may be a useful marker in high-risk individuals or when combined with other imaging or blood-based markers for pancreatic cancer. Furthermore, this work suggests that a cachexia-like phenotype is present early in pancreatic cancer development in humans and understanding this biology may allow more specific biomarkers to be identified.

We discuss strategies for using skeletal muscle wasting in pancreatic cancer surveillance on page 14 (paragraph 3) of the revised manuscript:

“Given the potential for tissue changes to occur with other conditions, general population screening for pancreatic cancer would need to consider multiple factors in combination and solely evaluating adipose tissue and skeletal muscle is unlikely to be a fruitful strategy. Thus, future studies will need to evaluate high-risk populations, define thresholds of skeletal muscle change over time with appropriate sensitivity and specificity for predicting pancreatic cancer diagnosis, and account for the need to standardize tissue measurements by age, sex, and race.”

Reviewer #3 (Remarks to the Author):

Very nice study. Strengths include the many prediagnostic scans, from two sites, the CT algorithm for quantifying the tissue compartments, and the data available for the case/control analysis. No significant weaknesses.

We thank the Reviewer for these comments.

The authors should comment on the small % of muscle mass that occurs in some very old adults just to put into context that this % is much smaller and less common than the loss seen in prediagnostic PDAC.

We agree with the Reviewer that muscle loss can occur in older adults as they age, and as the Reviewer notes, this muscle loss is generally less than that seen prior to a pancreatic cancer diagnosis. In the current study, we accounted for differences in tissue areas by age, sex, and race by standardizing all tissue areas to a large reference population of >12,000 individuals who underwent an outpatient CT scan. We agree that this is an important aspect of the current study and the need to account for differences in tissue areas as people age is important to mention in the Discussion section.

In response to the Reviewer's comment, we now state in the Discussion section on page 15 (paragraph 2) of the revised manuscript:

"These individuals originated from two large, geographically distinct health systems, and tissue areas were standardized to a large reference set of more than 12,000 outpatient CT scans to minimize variability of raw tissue measurements in relation to age, sex, and race. Standardization of tissue areas is of particular importance given these areas vary with age, sex, and race, as we have shown previously,[14] and tend to slowly decline in older adults."

The sentence below should be clarified to make it clear it is referring to the pilot study (Ref 13) cited in the prior sentence.

"In this study evaluating raw tissue areas on CT imaging..."

To something like...

In that study (Ref 13) evaluating raw tissue areas on CT imaging...

In the revised manuscript, we have now edited the text as suggested in the Discussion section on page 13 (paragraph 2):

"In that prior study evaluating raw tissue areas on CT imaging, subcutaneous adipose tissue appeared to decline first, followed by loss of skeletal muscle and visceral adipose tissue. [13] In contrast, in our study population evaluated with age-, sex-, and race-standardized tissue measurements..."

REVIEWERS' COMMENTS

Reviewer #2 (Remarks to the Author):

The authors have adequately addressed the reviewer concerns.

Reviewer #3 (Remarks to the Author):

Responses to reviewer comments are satisfactory.

Reviewer #4 (Remarks to the Author):

Thank you for your thoughtful responses to the documented critiques. All concerns have been addressed and modified where necessary.